# Transcriptome Analysis of In Vitro Fertilization and Parthenogenesis Activation during Early Embryonic Development in Pigs

**DOI:** 10.3390/genes12101461

**Published:** 2021-09-22

**Authors:** Xin Li, Cheng Zou, Mengxun Li, Chengchi Fang, Kui Li, Zhiguo Liu, Changchun Li

**Affiliations:** 1Key Laboratory of Agricultural Animal Genetics, Breeding and Reproduction of the Ministry of Education and Key Laboratory of Swine Genetics and Breeding of the Ministry of Agriculture, Huazhong Agricultural University, Wuhan 430070, China; xinerdi1234@163.com (X.L.); zzcy873704025@163.com (C.Z.); 18699301319@163.com (M.L.); fangchengchi22@aliyun.com (C.F.); 2The Cooperative Innovation Center for Sustainable Pig Production, Wuhan 430070, China; 3Agricultural Genome Institute at Shenzhen, Chinese Academy of Agricultural Sciences, Shenzhen 518124, China; likui@caas.cn; 4State Key Laboratory of Animal Nutrition, Key Laboratory of Animal Genetics Breeding and Reproduction of Ministry of Agriculture and Rural Affairs of China, Institute of Animal Sciences, Chinese Academy of Agricultural Sciences, Beijing 100193, China

**Keywords:** pig, parthenogenesis activation, in vitro fertilization, lincRNAs, imprinted genes

## Abstract

Parthenogenesis activation (PA), as an important artificial breeding method, can stably preserve the dominant genotype of a species. However, the delayed development of PA embryos is still overly severe and largely leads to pre-implantation failure in pigs. The mechanisms underlying the deficiencies of PA embryos have not been completely understood. For further understanding of the molecular mechanism behind PA embryo failure, we performed transcriptome analysis among pig oocytes (meiosis II, MII) and early embryos at three developmental stages (zygote, morula, and blastocyst) in vitro fertilization (IVF) and PA group. Totally, 11,110 differentially expressed genes (DEGs), 4694 differentially expressed lincRNAs (DELs) were identified, and most DEGs enriched the regulation of apoptotic processes. Through cis- and trans-manner functional prediction, we found that hub lincRNAs were mostly involved in abnormal parthenogenesis embryonic development. In addition, twenty DE imprinted genes showed that some paternally imprinted genes in IVF displayed higher expression than that in PA. Notably, we identified that three DELs of imprinted genes (*MEST*, *PLAGL1*, and *DIRAS3*) were up regulated in IVF, and there was no significant change in PA group. Disordered expression of key genes for embryonic development might play key roles in abnormal parthenogenesis embryonic development. Our study indicates that embryos derived from different production techniques have varied in vitro development to the blastocyst stage, and they also affect the transcription level of corresponding genes, such as imprinted genes. This work will help future research on these genes and molecular-assisted breeding for pig parthenotes.

## 1. Introduction

Pig (*Sus scrofa*) is one of the most important domesticated animals and a well-suited biomedical model for human disease because of its similarities in anatomy and physiology with humans [1,2,3,4]. Parthenogenesis is a common model of asexual reproduction that does not involve males, and is common in lower species, but it has been reported only in few vertebrate species (approximately 0.1%) [5]. Embryos derived from parthenogenesis activation (PA) are valuable for studies on porcine embryonic stem cells (pESCs) and gene imprinting with few ethical issues [6,7,8]. However, embryos generated from PA without paternal genomes result in embryonic death in mammals. Mammalian PA embryonic death occurs at 12, 21, 28, and 30 days of gestation in rabbits, sheep, pigs and cattle, respectively [9]. It is well known that the improvement of parthenogenesis activation technology in pigs must be based on the understanding of its mechanism and characteristics. However, there are still many problems in parthenogenesis activation technology.

In the current study, some biological processes seem to contribute to the defective development of paternal genomes, such as defective placentation [10,11], the failure of X-inactivation in the trophectoderm [12], lack of a heterochromatin halo around nucleolar precursors [13], DNA damage and mitochondrial dysfunction [14], differences in epigenetic modifications, and so on [15]. These studies have guided the direction of parthenogenesis activation failure, but a paucity of information is available regarding dynamics of transcriptional regulation during the early embryonic development of parthenogenetic activation. By applying quantitative proteomics combined with an RNA sequencing approach of the oocytes and embryos from eight stages in buffalo, a unique differentially expressed gene set was identified, and they found that maternal expression of some proteins possibly plays a role in the formation of cellular junctions first after parthenogenesis activation [16]. In pigs, the RNA-seq of embryonic disc (ED) and trophectoderm (TE) cells from in vivo fertilization, in vitro fertilization (IVF), parthenogenetic oocyte activation (PA), and somatic cell nuclear transfer (SCNT) embryos was performed to observe gene expression profiles of various embryo types during peri-attachment development [17]. However, we know little about dynamic transcriptome during early embryonic development in IVF or PA.

LincRNAs are a class of intergenic transcripts that are greater than 200 nt in length and have limited protein-coding potential. These lncRNAs have been increasingly recognized as an important regulatory factor of gene expression and act as a molecular platform that associates different domains with DNA, RNA, or proteins [18,19,20] in various biological processes such as gene regulation, maintenance of pluripotency, and transcriptional regulation [21,22]. Through whole transcriptome sequencing, a growing list of lincRNAs that were significantly involved in embryonic morphogenesis and development have been identified in mice, rabbit, and pig [23,24,25]. For example, several lincRNAs, such as H19, p21, and Xist, have been characterized by their functional relevance to embryogenesis [26,27,28]. However, most lincRNAs and their function remain unclear in vitro mature embryos by parthenogenesis activation or fertilization.

Genomic imprinting is an epigenetic process regulated by germline-derived DNA methylation that leads to parental allele-specific gene expression and, thus, non-equivalent and complementary function of parental genomes [29]. Genomic imprinting not only plays critical roles in fetal development and placenta and brain function [30], but also poses an epigenetic barrier to parthenogenesis in sexual organisms. In bovine, some imprinted genes that were identified in vivo produced oocytes and embryos using RNA-seq, such as *MEST, PLAGL1, CDKN1C, IGF2R, SGCE, PEG10, PEG3, H19, XIST*, and so on [31]. As mentioned above, *Xist*, as an important imprinted gene, can directly affect the development of porcine parthenogenetic embryos. In addition, the expression of most maternally imprinted genes is increased, whereas that of paternally imprinted genes is significantly reduced in parthenogenetic embryos [32,33]. However, the roles of most imprinted genes in the early development of pig parthenogenesis activation embryos still needs more in-depth research.

For our analysis, we focused on the differentially expressed analysis between in vitro fertilization and parthenogenesis activation embryos and the functional prediction of putative lincRNAs. Based on functional enrichment information, we further identified some potential genes and lincRNAs that had important roles in the failure of pig parthenogenesis activation embryos, such as MEST_lincRNA, PLAGL1_lincRNA, and DIRAS3_lincRNA. Notably, those lincRNA corresponding genes have previously been reported as imprinted genes in pigs. Hence, our study provides an in-depth understanding of the dynamics of transcriptional regulation during early embryonic development and some insights into putative lincRNAs of imprinted genes that may improve our understanding of parthenote failure in porcine.

## 2. Materials and Methods

### 2.1. Animal Ethics Statement 

Animal slaughtering, sample collection, and experiments related to animal work in this study were carried out according to the pre-approved guidelines of the Institutional Animal Care and Use Committee (IACUC) at the Institute of Animal Sciences, Chinese Academy of Agricultural Sciences (IAS2020-25). The experimental protocol in our study was approved by the Institutional Animal Care and Use Committee (IACUC) at the Institute of Animal Sciences, Chinese Academy of Agricultural Sciences on 1 March 2020.

### 2.2. Collection and Culture of Pig Oocytes and Embryos

#### 2.2.1. Oocyte Collection

In this study, porcine ovaries were obtained from a slaughterhouse and transported to the laboratory within 3 h while maintained at 38 °C. After washing ovaries three times in warm physiological saline solution supplemented with streptomycin and penicillin, an 18-gauge needle attached to a 10-mL disposable syringe was used to aspirate the follicular fluid from 3 to 6 mm follicles. Cumulus oocyte complexes (COCs) with uniform cytoplasm and several layer of cumulus cells were selected, and 3–5 layers of these cumulus cells were selected and then washed twice in wash buffer. In vitro maturation, conducted in 4-cell dishes in TCM-199 culture medium, included incubation at 38.5 °C with 5% CO_2_ in air, at maximum humidity, for 42–44 h. Mature oocytes were chosen with first polar body and uniform cytoplasm for the next step [34].

#### 2.2.2. In Vitro Fertilization

Groups of 30 denuded oocytes were washed three times in vitro fertilization (IVF) medium, and then placed in 100 µL drops of in vitro fertilization (IVF) medium covered with paraffin oil, which were held at 38.5 °C in an atmosphere of 5% CO_2_ in air for approximately 30 min, until the addition of spermatozoa. Cryopreserved semen pellets were thawed in warm Beltsville Thawing Solution (BTS), loaded on a 65–70% Percoll (E0414) gradient, and centrifuged 2000× *g* for 15 min. The pellet was re-suspended in warm BTS and centrifuged at 300× *g* for 10 min, and then the final pellet was re-suspended in the modified Tris-buffered medium. The semen suspension (50 µL) was added to a drop of IVF medium containing the oocytes. The oocytes were incubated with the sperm for 10 hr at 38.5 °C with 5% CO_2_ in air. Eighteen hours after fertilization, fluorescence stains (5 µg/mL H33342, 10–20 min) were used to observe prokaryotic formation. One female pronucleus, one male pronucleus, two polar bodies, and non-agglutinated sperm heads were used as the standard for normal fertilization, and then according to the type of pronucleus formation recorded the number of embryos of various types [34,35].

#### 2.2.3. Parthenogenetic Activation

Groups of 30 denuded oocytes were placed between 0.2-mm-diameter platinum electrodes 1 mm apart in activation medium. Activation was induced with two direct-current (DC) pulses of 1.2 kV/cm for 40 μs on a BTX Elector-Cell Manipulator 200 (BTX, San Diego, CA, USA) according to the experimental design. The medium used for activation was 0.3 M mannitol, supplemented with 1.0 mM CaCl_2_, 0.1 mM MgCl_2_, and 0.5 mM Hepes. The orientation of oocytes and polar bodies was not vertical to platinum wire electrodes during electrical activation. The parthenogenesis activation (PA) embryos were washed and exposed to NCSU medium with 5 μg/mL cytochalasin B for 4 h to inhibit second polar body extrusion, then cultured in 150 µL NCSU medium covered with mineral oil in a 96-well culture plate. The embryos were incubated at 38.5 °C with 5% CO_2_ in air [36].

Single cells of porcine MII (meiosis II) oocytes, zygote (12 h), morula (5–6 d), and blastocyst (6–7 d) stage derived from in vitro fertilization and parthenogenesis activation were collected as described previously and stored individually. The day of in vitro fertilization or electrical activation is set to day 0. All embryo samples were placed on dry ice immediately after collection and stored at −80 °C.

### 2.3. Preparation of cDNA and Transcription Profiling 

Total RNA was extracted from 21 single cells using an RNA extraction kit (SMART-Seq^®^ v4 Ultra™ Low Input RNA Kit (Takara, San Jose, CA, USA) according to the manufacturer instruction. RNA purity was qualified using NanoDrop ND 2000 spectrophotometer at 260 and 280 nm (Thermo Fisher Scientific, Wilmington, MA, USA), and RNA integrity was verified using an Agilent 2100 Bioanalyzer (Agilent Technologies, Palo Alto, CA, USA). The OD260/280 ratios of all the samples were greater than 1.8, and the RIN values were greater than 7. According to the published protocol [37], Cdna preparation and amplification were performed. AMPure XP beads (Beckman Coulter, Brea, CA, USA) were used for PCR products purification. Before library preparation, the Cdna quantity and quality were assessed by Qubit and agarose gel electrophoresis. Products were purified (AMPure XP system, Beckman Coulter) and libraries were sent to Novogene for quality assessment and sequencing. The sample quality was assessed using NanoDrop, agarose gel electrophoresis, and the Agilent 2100 Bioanalyzer (Agilent Technologies). The sequencing process was performed with 150 bp paired end on an Illumina HiSeq 2500 instruments (Novogene, Tianjin, China).

### 2.4. Quality Control (QC) and Transcriptome Assembly 

The raw reads were cleaned by filtering the adapter and low-quality reads by Trimmomatic (version 0.3.2) [38]. First, the adapters were removed; then, low quality reads (mismatched bases that account for more than 50% of a read) were removed, and the reads in which the average quality of four continuous bases was <15 were discarded. Then, the clean reads were mapped to the pig reference genome (Sus scrofa 11.1) by HISAT2(version 2.0.1). According to different database construction methods, we selected the corresponding parameters “known-splicesite-infile” and got the sam file containing all mapping readings for each sample. Using samtools (version 1.3.1), the sam files were processed and converted into sorted bam files with parameters “view –H” and “sort –o”. Finally, for facilitating transcript assembly and quantification, all bam files were assembled into one complete GTF file with the default parameters and merge function of StringTie (version 1.3.5) [39]. 

### 2.5. Expression Calculation and Differentially Expressed Analysis

Read count tables were generated from binary sequence alignment/map (BAM) files using Htseq software. The value for fragments per kilobase of exon per million fragments mapped (FPKM) [40] was calculated to estimate gene expression under each sample following the rules below:$$\mathrm{FPKM}=\frac{\mathrm{Read}\;\mathrm{count}\times 10^{6}\;}{\mathrm{Total}\;\mathrm{reads}\;\mathrm{mapped}\;\mathrm{to}\;\mathrm{genome}\;\times \mathrm{Gene}\;\mathrm{lenght}\;\left(\mathrm{kb}\right)}$$

The DESeq2 software package was used to calculate differences in gene expression of IVF and PA and the same stage between IVF and PA. The differentially expressed transcripts with log2 (FoldChange) greater (less) than 1 (−1) and an adjusted *p*-value less than 0.05, are considered to be significant [39]. 

### 2.6. Identification and Function Analysis of lincRNAs

By using the assembled GTF files, StringTie software was able to estimate the expression levels of genes and transcripts in all samples for subsequent studies with the parameters “-e” and “-B” [39]. And the assembled transcriptome was then filtered to obtain the putative lincRNAs. Our pipeline for lincRNA identification as shown in Appendix A was based on the way described in our previous study [41,42].

#### 2.6.1. Neighboring Gene Analysis

For each differentially expressed lincRNA (DEL) locus, the nearest protein-coding neighbor within <100,000 bp was identified by BEDtools (version 2.17.0) [43,44,45], and these neighboring genes were expressed in at least one sample. This resulted in a list of lincRNA protein-coding loci pairs. Pearson correlation was used to explore the expression-based relationship between these pairs. Ensemble was used to convert the protein-coding gene id into a human gene id for annotation, visualization, and integrated discovery (DAVID) [46,47]. Then, we performed GO analysis and KEGG pathway analysis on all expressed protein-coding genes of DELs. Terms with a *p*-value < 0.05 were considered statistically significant.

#### 2.6.2. Weighted Gene Co-Expression Network Analysis

Co-expression networks were constructed by WGCNA (version 3.5.2) [48] package in the RStudio environment (version 3.5.3) using two parts of genes (differentially expressed putative lincRNAs and differentially expressed protein-coding genes). A signed weighted correlation network was constructed by creating a matrix of pairwise Pearson correlation coefficients, and module detection was used by the step = by-step signed network construction with a soft threshold of power = 18. Then, the topological overlap distance calculated from the adjacency matrix was then clustered with the average linkage hierarchical clustering. We retrieved the protein-coding genes that co-expressed with lincRNAs in each module, then GO enrichment and pathway analysis was performed on them. The minimum module size was set to 50 to ensure a qualified number of genes for further analysis. For each module, we defined the first principal component as the eigengene according to WGCNA terminology. To detect the relationship between modules and seven development stages, we defined a vector to encode these seven development stages. Then, we correlated this vector with the eigengenes of each module, and a higher correlation indicated that the module was related to the corresponding development stage.

#### 2.6.3. Identification of Hub LincRNAs

Two modules were finally left for identifying hub lincRNAs, and we calculated the connectivity of each gene based on their intra-module connectivity. LincRNAs with top 5% intra-module connectivity were defined as hub lincRNAs, networks of which were displayed graphically using Cytoscape (version 3.7.0) software. The correlation between hub lincRNAs and co-expression protein-coding genes must be larger than 0.9.

### 2.7. Gene Ontology and Pathway Analysis

We performed DAVID analysis by running queries for each protein-coding gene against the DAVID database. Because the annotation for the genes in *Sus scrofa* 11.1 was relatively limited, we used the BIOMART system to abstract orthologous relationships between pig and human gene pairs and convert mapping information regarding gene identities between human ensemble and reference sequences.

### 2.8. Validation of RNA-Seq Data by Quantitative Reverse-Transcription PCR 

To validate the RNA-seq results, we perfORMed qRT-PCR with SYBR Green I-based Real-time Quantitative pcR (qRT-PCR) (CWBIO, Beijing, China, CW0957). Ten pairs of primers for qRT-PCR were designed using Primer 5 program (Appendix A). ThE 18s rRNA served as the endogenous control gene in the Roche LightCyler 480 system (Roche, Mannheim, Germany) [49]. First, THe cDNA (500 ng/µL) were used as templates, which were obtained by reverse transcription and were diluted five times with RNase-Free H_2_O [50]. Then, we performed qRT-PCR according to the instructions, and the reaction system was shown in Appendix A. Three replicates and no template controls were set for each reaction. Finally, the qRT-PCR data were analyzed using the delta Ct ([Delta] [Delta]Ct) method. We used the trend of gene expression during total stages to judge whETHer qRT-PCR results were in accordance with the RNA-Seq results.

Venn diagram and heat map in our study were produced by Tbtools [51]. Histogram, line chart, density map, box plot, bubble chart of functional enrichment, and WGCNA were produced in the RStudio environment (version 3.5.3).

## 3. Results

### 3.1. Overview of Sequencing Data

In vitro embryo development of IVM (In vitro maturation) oocytes after IVF or PA was described in Table 1. Approximately 32.14% of cleavage embryos derived from PA developed to the blastocyst stage, while approximately 26.47% of cleavage embryos derived from IVF developed to the blastocyst stage. The embryos derived from different production techniques may have varied in vitro developmental potential, which is consistent with a previous report [17]. Furthermore, we investigated the difference of transcription between them through performing RNA sequencing of 21 single cells from three stages in IVF and PA. 

With RNA-seq transcriptome analysis, we obtained a total of 3484.4 million raw reads with an average of ~166.0 million reads per sample. After filtering adapter and low quantity reads, we detected 2303.2 million clean reads with 110.0 million reads per sample. Each RNA-seq dataset was separately aligned to the *Sus scrofa* (11.1) genome by using HiSat2. The result showed that most of the samples had a comparative high alignment rate of over 85% and a unique mapping rate range of 55–75%, except the PA_M_3 sample (Table 2). After reconstructing the transcriptome for each group and merging the 21 assembled transcripts into a non-redundant transcriptome, we counted the number of reads in each sample by using HTseq-count. Genes with fpkm ≥ 0.1 in at least three samples were considered the expressed genes. Totally, we identified 15,322 known protein-coding genes accounting for 59.2% of annotated porcine genes (25,880) in all stages (Appendix A). In the IVF or PA group, different genes were specifically expressed at different stages, and, notably, most genes were specifically expressed in the morula stage of the PA group (Appendix A). For each stage, the Pearson correlation analysis on repeat samples was performed, and the results were shown in Appendix A. Intraclass correlation coefficients of MII, IVF_Zygote, PA_Zygote, PA_Blastocyst stage were more than 0.9, while that of IVF_M, IVF_B, PA_M showed poor intraclass repeatability (0.39–0.91). This might have been led by uneven sample quality and poor sequencing quality. In the follow-up analysis, average expression was used to represent the transcription level of the stage. 

### 3.2. Differential Expression Analysis of mRNAs in Adjacent Stages, IVF, and PA

First, we normalized the transcript expression levels to the FPKM values by using HTSeq count. We then performed cluster analysis by Hierarchical Clustering (hclust) from R packages [52] with the FPKM of these 21 samples in all stages, and we found mature oocytes and zygote embryos clustered closely together but away from the morula and the blastocyst, indicating that the one-cell stage exhibited a distinct transcriptome pattern (Figure 1A). Then, we conducted differential expression analysis for IVF and PA samples by using DESeq2 to explore the early embryonic development-related genes. We totally identified 11,110 differentially expressed genes (DEGs) in IVF and PA (Appendix A). By comparing the number of DEGs in adjacent stages, we found the largest number of DEGs occurred in the blastocyst stage in IVF and morula stage in PA (Appendix A), indicating that embryos of the IVF and PA group had undergone a relatively recent burst of transcriptome after activation. Pathway enrichment analysis with the DEGs of these two stages both showed that DEGs in IVF or PA were mainly involved in transcriptome, cell division, cell proliferation and differentiation, energy metabolism, biosynthesis, and embryo development related-biological pathways. Different to IVF, some DEGs were significantly involved in cell-cell adhesion, cell cycle, and the regulation of apoptotic processes in the PA group (Appendix A). 

We further performed cluster analysis of all DEGs in IVF and PA by using the short time-series expression miner (Stem) software [53]. The functional annotation and enrichment analysis in each group for the nine clusters were estimated with *p*-value < 0.01 in IVF and PA, respectively (Figure 1B, Figure 2). These nine clusters of IVF and PA were significantly related to early embryonic development processes, such as transcription, cell cycle, and cell proliferation and differentiation. However, compared with those in IVF, many pathways related to the regulation of the apoptotic process occurred in the DEGs of parthenogenesis-activated embryos including *AIMP2, BIRC5, BNIP1*, and *BOK* [54,55,56,57].

By comparison, in the same stage between IVF and PA, we identified 133 (66 upregulated, 67 downregulated), 3667 (938 upregulated, 2729 downregulated), and 374 (211 upregulated, 163 downregulated) DEGs in the zygote, morula, and blastocyst stages, respectively. Then, we performed GO enrichment analysis of these DEGs and found some pathways related to basic cell development and metabolic processes in the morula stage, such as regulation of transcription and translation, energy metabolism, and cell proliferation and differentiation. We also found other pathways related to the apoptotic process in the morula stage, including regulation of the apoptotic process, extrinsic apoptotic signaling pathway, and regulation of cell death. In the blastocyst stage, most DEGs participated in transcription, cell proliferation and differentiation, cell cycle, and apoptotic-related processes (Appendix A).

### 3.3. Identification and Function Prediction Analysis of lincRNAs

Here, we filtered transcripts according to our recently updated pipeline to identify putative lincRNAs (Appendix A). Then, we obtained 15,070 putative lincRNA transcripts produced by 7185 gene loci, which were distributed in all chromosomes (Appendix A). Besides, 6113 novel transcripts of these 15,070 lincRNA transcripts had no overlap with currently annotated transcripts (Appendix A). Moreover, based on the comparison of the basic characterization between lincRNAs and protein-coding genes, lincRNAs showed shorter transcript length, longer exon length, smaller exon number, and lower expression (Appendix A), which were consistent with previous reports [58]. In addition, we performed differential expression analysis of lincRNAs in adjacent stages (Appendix A). A large number of DEGs and DELs both occurred at the same stage (Appendix A), indicating that the DELs might cooperate with the DEGs and thereby follow their expression trends.

The prediction of the functions of lincRNAs remains challenging due to their lack of annotation and low expression levels. Generally, the function of lincRNAs can be predicted via cis-acting manner (proximity regulation) and trans-acting manner (correlation prediction). LincRNAs could act in cis to regulate their neighboring genes [59]. Correlation-based approaches have also been used to infer the function of lincRNAs [60]. To explore the potential cis-acting of lincRNAs, we performed GO enrichment analysis of expressed protein-coding genes nearby the lincRNAs (<100 kb) (Appendix A). We found that these neighboring protein-coding genes of pig lincRNAs were mostly enriched in biological processes or pathways related to transcription, protein modification, and cell development, such as transcription from RNA polymerase II promoter, nature killer cell activation involved in immune response, positive regulation of peptidyl-seine phosphorylation of SAT protein, regulation of cell cycle arrest, and cell–cell adhesion (Figure 3) (Appendix A). 

Moreover, we performed a transcriptome-wide weighted gene co-expression network analysis (WGCNA) to infer the potential roles of lincRNAs in IVF and PA. We identified 15 co-expression modules with sizes ranging from 74 to 3841 (mean: 647; median: 273). Then, we filtered some modules by stages specific (correlation ≤ 0.6, *p*-value ≤ 0.05) (Figure 4A) and numbers of lincRNAs per each module (*n* < 1). In total, six modules (blue, green, green–yellow, pink, purple, and turquoise) were left, and then GO enrichment analysis of protein-coding genes in these six modules was performed (Appendix A). Most of the modules were enriched for transcription regulation, cell cycle, cell proliferation, and metabolic processes. This finding suggested that the lincRNAs in these modules might play similar roles in early embryonic development [61,62,63] (Figure 4B).

Next, we selected these six modules for hub lincRNA analysis to further clarify the function of hub lincRNAs in two modules. We measured the intramodular connectivity (also named weight) of each gene by WGCNA to select the top 5% lincRNAs as hub lincRNAs in each module. Totally, we identified five hub lincRNAs in the green module and three in the green–yellow module, respectively. Two co-expressed networks between these hub lincRNAs and protein-coding genes were constructed (Appendix A). In module green, the enrichment pathways of protein-coding genes mainly involved methylation-dependent chromatin silencing, transcription from polymerase II promoter, and mRNA. Some of these genes had a crucial role in abnormal parthenogenesis embryonic development, such as *PPP2CA*, *NANOG*, *DMXL2*, *GATM*, *API5*, and *WDR36* [64,65,66,67,68,69]. Other genes, such as *DDX5*, *POLG2*, *SPATA22*, *CKS2*, *ORC4*, and *SPC25*, had an important role in transcriptome and cell development in early embryonic development [70,71,72,73,74,75]. In module green–yellow, the enrichment pathways of protein-coding genes mostly included multicellular organism development and regulation of transcription. The *TNFAIP6*, *DIS3L2*, *PAK5*, *DDB1*, and *KDM8* genes in this module had important roles in cell proliferation in early embryonic development [76,77,78,79,80]. These related genes shared a high correlation (r > 0.9) and a highly similar expression trend with hub lincRNAs. Hence, these five hub lincRNAs in module green on abnormal transcriptome and basal metabolism and the three hub lincRNAs in module green–yellow might participate in cellular development by regulating cell proliferation. Although the detailed mechanisms of how these hub lincRNAs are involved in embryonic development are quite limited, these lincRNAs would serve as ideal candidates for further functional studies.

### 3.4. Identification of Imprinted Genes in Pig

As we know, incomplete epigenetic reprogramming of imprinted genes is a barrier to parthenogenetic birth in mammals [81]. Imprinted genes can act directly on the fetus by influencing cellular proliferation or apoptosis, and they can also affect fetal growth by influencing the flux of maternal nutrients through the placenta [82]. Although some imprinted genes have been reported in pigs, there is still a considerable amount of information missing. In IMPRINTED GENE DATABASES, we found fifty-five pig imprinted genes had been reported, and twenty-six of them were quantified in this study by RNA-seq. Twenty imprinted genes were differentially expressed (thirteen paternally and seven maternally expressed; 17 DEGs and three DELs). Some imprinted genes showed significantly higher expression in IVF compared to PA group, especially paternally imprinted genes, such as *PEG10* (IVF_M vs. PA_M, log2(FC) = 2.84), *KBTBD6* (IVF_B vs. PA_B, log2(FC) = 5.82), and *INPP5F* (IVF_M vs. PA_M, log2(FC) = 1.433) (Figure 5A). *MEST*, *PLAGL1*, and *DIRAS3* were also three putative DELs. Interestingly, 11 of 20 DE imprinted genes were identified as differentially expressed imprinted genes in pig parthenogenetic fetuses, such as *MEST*, *PLAGL1*, *PEG3*, *PRIM2*, *PEG10*, *DIRAS3*, *SGCE*, *PON2*, *OSBPL1A*, and *AMPD3* [82]. Furthermore, a qRT-PCR for *PHLDA2*, *PEG10*, *DIRAS3*, and *INPP5F* validated that expression of imprinted genes in IVF were higher than in the PA group (Figure 5B). 

### 3.5. Quantitative Reverse-Transcription PCR Validation of Transcriptome Sequencing Results

We randomly selected eight genes (four protein-coding genes and four lincRNAs) from the comparative gene expression data, and we then evaluated their expression in different stages using SYBR Green I-based Real-time Quantitative PCR (qRT-PCR). The primer sequences for all target genes were listed in Appendix A. All selected genes (LDHB, HSPE, NUP35, C14orf166, MSTRG.12993, MSTRG.53456, MSTRG.58641, and MSTRG.341) showed a statistical consistency with our sequencing data (Figure 5B), thereby further improving our research reliability.

## 4. Discussion

We developed a platform for mRNA sequencing during pig early embryonic development in vitro fertilization and parthenogenesis activation. These results provided a comprehensive framework of transcriptome landscape for 21 single cells of mature oocyte and three typical stages (zygote, morula, and blastocyst). Our major findings include the following:

According to the result of differential expression analysis, IVF and PA had a significant difference on the most active stage of transcription. Embryos of the IVF and the PA group underwent a relatively recent burst of transcriptome in the blastocyst stage (IVF) and morula stage (PA), respectively. As previously reported, the development rate of parthenogenesis activation oocyte cleavage was advanced by at least 6 h compared with in vitro fertilized embryo in the actual experiment in buffalo, indicating that the time of compaction during early development of parthenogenesis activation embryo was earlier than the fertilized eggs [16]. It seems very likely that the difference in the most active stage of transcription in IVF and PA is led by the advanced cleavage of PA oocytes.

Through functional enrichment analysis of DEGs, we found that the DEGs of the PA group were not only enriched in early embryonic development related-biological pathways, but also in cell-cell adhesion, cell cycle, the regulation of apoptotic processes, and so on. Additionally, differentially expressed, and functional enrichment analysis, of the same stages between IVF and PA showed that energy metabolism, cell proliferation and differentiation, regulation of apoptotic process, extrinsic apoptotic signaling pathway, and regulation of cell death were significantly enriched in the morula stage, while cell proliferation and differentiation, cell cycle, and apoptotic-related process were enriched in the blastocyst stage. An increase in apoptosis resulted in embryo loss and lower developmental competence of in vitro–cultured embryos [83]. The pro-apoptotic profile is more pronounced in parthenogenetic embryos than in IVF embryos in bovine, and PA embryos also showed less cellular metabolism which leads to lower health of embryos [5,84]. Thus, we inferred that disordered expression of DEGs related to apoptotic processes can very likely account for the development failure in PA embryos, such as *AIMP2*, *BIRC5*, *BNIP1*, and *BOK*. As previous shown, AIMP2 played pivotal roles in the regulation of cell proliferation and death, and mice lacking it would result in abnormal embryonic development [54].

A large number of lincRNAs can be found in mammalian genomes, and their exact number may be equal to or even surpass the number of protein-coding genes. We performed a comprehensive identification and functional prediction of putative lincRNAs via cis-acting (proximity regulation) and trans-acting (co-expression networks analysis) manner. Here, we identified 7185 putative lincRNAs (6113 novel putative lincRNAs), which have broadened the pig lincRNA annotation. Moreover, through similar functions among neighboring genes, most of the protein-coding genes near lincRNA (<100 kb) were enriched in transcription, protein modification, and cell development process, and thus we inferred lincRNAs in our study might largely have similar roles. After constructing co-expression networks between DELs and DEGs, we obtained six modules that DELs and DEGs in each module had the same expression trend, and most of the modules were enriched for transcription regulation, cell cycle, cell proliferation, and metabolic processes. Notably, stage-specific module green and green–yellow (module green: PA_M; module green–yellow: IVF_Z) were possibly related to abnormal parthenogenesis embryonic development and cell proliferation in early embryonic development, respectively. In module green, some of these genes had a crucial role in abnormal parthenogenesis embryonic development, such as *PPP2CA*, *NANOG*, *DMXL2*, *GATM*, *API5*, and *WDR36*. In module green—yellow, *TNFAIP6*, *DIS3L2*, *PAK5*, *DDB1*, and *KDM8* genes in this module had important roles in cell proliferation in early embryonic development. However, no significant differences were observed in vitro embryo development of IVM oocytes after IVF or PA. It suggested that parthenogenesis activation oocytes did not impair in vitro pre-implantation development to the blastocyst stage, but lead disordered expression of corresponding genes, and then affected parthenogenesis embryo failure.

Parthenogenesis is a process in which zygotes are produced without sperm presence. Due to a lack of paternal genes, parthenogenetic embryos always failed to develop into a normal individual in mammals [15]. Genomic imprinting is controlled epigenetically by parental-specific DNA methylation imposed during the gametogenesis and induces gene expression bias in both chromosomal homologs, which is important for the regulation of normal embryonic development [85]. In total, 20 of 55 imprinted genes displayed significantly differential expression in this study. Three of those were observed in putative DELs, MEST_lincRNA, PLAGL1_lincRNA, and DIRAS3_lincRNA. Expression of MEST in this study (IVF_M vs. IVF_B: log2(FC) = 5.7; PA_M vs. PA_B: log2(FC) = −3.1), was consistent with previous reports [86]. The log2(FC) of PLAGL1 and DIRAS3 are 8.3 and 4.6 between IVF_M and IVF_B, while there was no difference in PA group. Interestingly, mesoderm specific transcript (MEST) is necessary for pig normal early embryonic development, but there is no expression of MEST-1C in PA fetuses. The hypermethylation of MEST-1C was observed in PA samples, which may be contributed to developmental retardation in pig parthenogenetic fetuses [86]. In 2006, PEG10 was confirmed as an essential role in mouse early parthenogenetic development [87]. In humans, the expression of PLAGL1/ZAC1 is associated with reduced growth rates and intellectual disability. In mouse models, the expression of PLAGL1/ZAC1 displayed altered brain sizes and cellular defects [88]. In total, disordered expression of imprinted genes in pig may be the reason for parthenogenetic-activated embryonic failure. The three putative lincRNAs of imprinted genes mostly have important roles in pig parthenogenesis activation embryo failure, for example, as a key regulator in apoptosis process, cell cycle arrest, or cell proliferation. However, the exact characterization of those three lincRNA of imprinted genes remains to be further elucidated. In conclusion, our findings will help future research on molecular mechanisms of parthenogenesis activation failure and molecular-assisted breeding for pig parthenotes.

## Figures and Tables

**Figure 1 genes-12-01461-f001:**
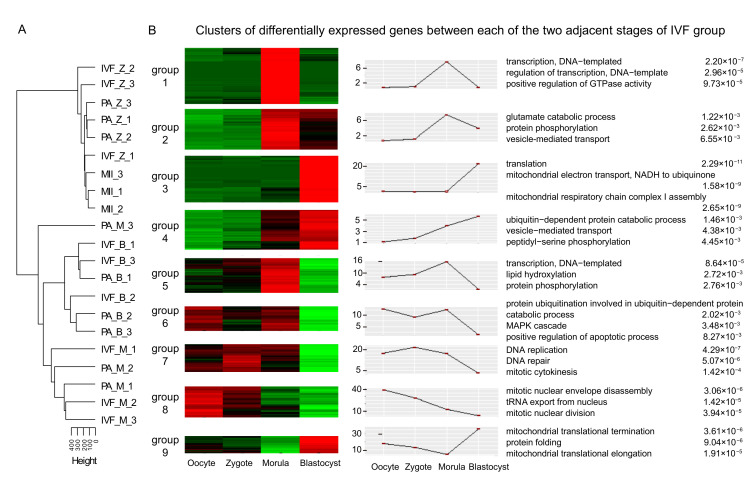
Unsupervised clustering of the expression profiles and differentially expressed genes (DEGs) between adjacent stages of pig embryos in vitro fertilization (IVF). (**A**) Unsupervised hierarchical clustering of the expression profiles; (**B**) Clusters of DEGs between adjacent stages of pig embryos in vitro fertilization (IVF). Gene average log transformed expression values, top GO terms, and corresponding enrichment *p*-values were listed.

**Figure 2 genes-12-01461-f002:**
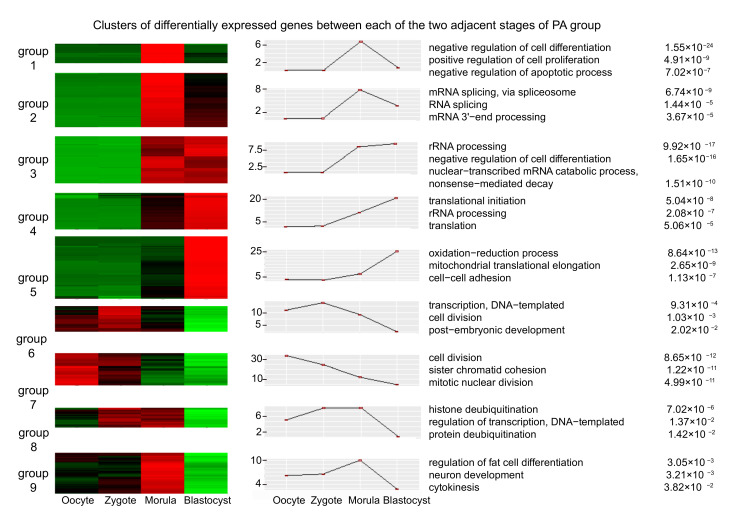
Clusters of DEGs between two adjacent stages of pig parthenogenesis activation (PA) embryos. Gene average log transformed expression values, top GO terms, and corresponding enrichment *p*-values were listed.

**Figure 3 genes-12-01461-f003:**
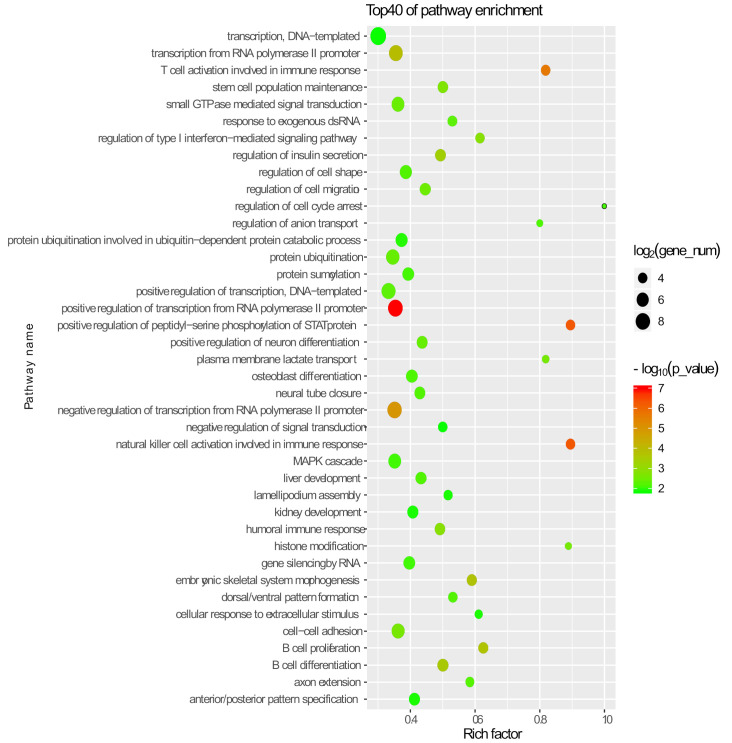
Top Go terms of its neighboring protein-coding genes.

**Figure 4 genes-12-01461-f004:**
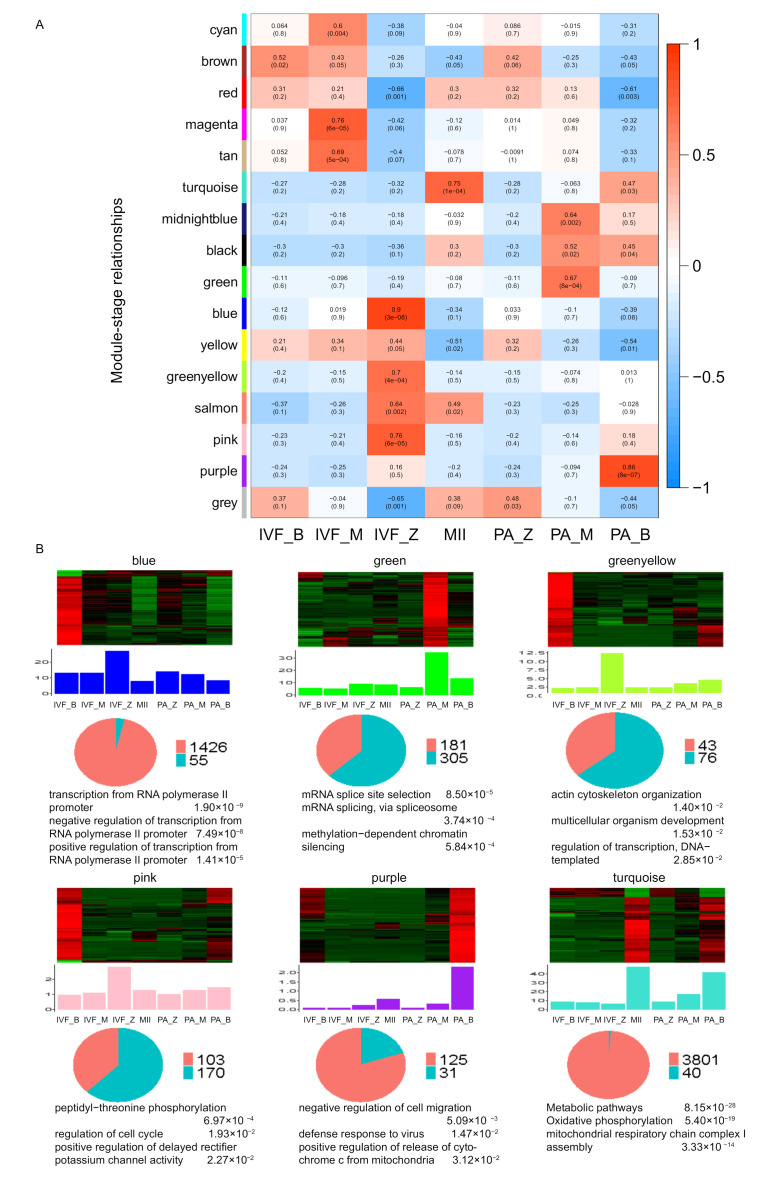
Module-stage correlation analysis and co-expression network analysis of differentially expressed novel lincRNAs and protein-coding genes. (**A**) Module-stage correlations and corresponding *p* values; on the left, different colors represent different modules; on the right, red indicates positive correlation, white indicates no correlation, blue indicates negative correlation; each cell contains the correlation and *p* value given in parentheses. (**B**) Co-expression networks of differentially expressed novel lincRNAs and protein-coding genes in six modules; top of each panel: heat maps for expression level of co-expressed genes in six modules. Red, increased expression; green, decreased expression. Middle of each panel: bar plots of the average expression of corresponding module eigengenes. Bottom of each panel: pie charts showing the abundance of lincRNAs and protein-coding genes and top Go terms of the latter within each module.

**Figure 5 genes-12-01461-f005:**
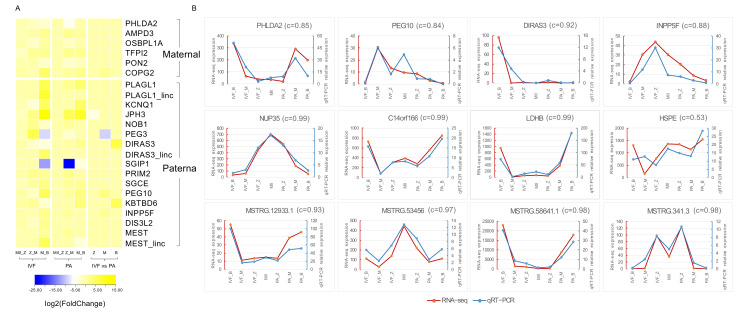
Differential expression analysis of imprinted genes and qRT-PCR validation of RNA-seq. (**A**) Heat map of DE imprinted genes. (**B**) qRT-PCR analysis of randomly selected imprinted genes, protein-coding genes, and lincRNAs. Red line represents average expression of RNA-seq; blue line represents average expression of qRT-PCR, and C value is the correlation between RNA-seq and qRT-PCR.

**Table 1 genes-12-01461-t001:** In vitro embryo development of IVM (In vitro maturation) oocytes after IVF or PA.

Embryo Source	Total Embryos (0 h)	Cleaved Embryos (48 h)	Blastocysts (6–7 d)
IVF	125	108 (86.38%)	33 (30.56% ^a^ –26.40% ^b^)
	305	183 (60.00%)	49 (26.77% ^a^ –16.07% ^b^)
	129	86 (66.67%)	19 (22.09% ^a^ –14.73% ^b^)
PA	55	46 (83.64%)	15 (32.60% ^a^ –27.27% ^b^)
	47	43 (91.49%)	11 (25.58% ^a^ –23.40% ^b^)
	49	34 (69.39%)	13 (38.24% ^a^ –26.53% ^b^)

Cleaved embryo yield percentages are calculated on the basis of the cleaved embryos (^a^) and on the basis of the initial number of oocytes (^b^).

**Table 2 genes-12-01461-t002:** In vitro embryo development of IVM oocytes after IVF or PA.

Sample Name	Raw Reads	Clean Reads	Clean Bases	Error Rate (%)	Q20 (%)	Q30 (%)	GC Content (%)	Uniquely Mapping Rate	Overall Alignment Rate
IVF_Z_1	137468078	104586902	13.01 G	0.03	96.73	92.23	44.97	65.65%	92.07%
IVF_Z_2	142771986	100439624	12.43 G	0.02	97.53	94.89	44.98	74.25%	91.80%
IVF_Z_3	135512064	102006570	12.63 G	0.02	96.36	92.48	46.97	73.66%	94.41%
IVF_M_1	156959598	98837458	12.36 G	0.04	98.78	96.27	45.57	74.38%	95.23%
IVF_M_2	169026320	105937556	13.09 G	0.03	97.59	94.78	46.02	65.17%	92.75%
IVF_M_3	156859688	100002906	12.37 G	0.03	97.7	95.13	47.08	66.95%	92.47%
IVF_B_1	150759132	99269248	12.33 G	0.02	97.74	94.3	48.21	70.28%	94.86%
IVF_B_2	181545464	115480292	14.26 G	0.02	95.78	92.17	39.67	63.31%	92.36%
IVF_B_3	146705060	100166918	12.38 G	0.02	98.09	95.89	49.42	71.38%	93.21%
MII_1	139913758	104812570	13.01 G	0.03	95.55	90.93	45.49	76.37%	95.77%
MII_2	165517814	112417876	14 G	0.02	98.04	95.7	45.71	47.38%	82.04%
MII_3	176029002	129015192	16 G	0.02	97.92	95.13	45.12	69.03%	93.92%
PA_Z_1	136106014	100661316	12.5 G	0.03	95.95	91.54	46	69.40%	92.33%
PA_Z_2	150617128	109910972	13.67 G	0.02	97.88	95.13	45.91	74.20%	94.61%
PA_Z_3	146127012	99562818	12.38 G	0.02	97.69	94.81	46.23	75.18%	94.60%
PA_M_1	223196572	100931396	12.42 G	0.04	97.52	94.85	44.97	65.31%	91.52%
PA_M_2	188725788	101995272	12.54 G	0.03	96.72	93.15	42.28	56.11%	89.39%
PA_M_3	181377016	110631938	13.57 G	0.03	95.48	91.64	43.6	47.55%	82.89%
PA_B_1	197029468	123993046	15.39 G	0.02	98.76	97.11	47.02	76.96%	95.96%
PA_B_2	219332164	144578468	17.82 G	0.02	95.57	91.51	42.61	51.02%	85.29%
PA_B_3	182772226	138010990	17.07 G	0.02	95.79	91.62	44.59	55.11%	87.73%

MII: MII (meiosis II) oocyte. IVF_Z: in vitro fertilization zygote stage. IVF_M: in vitro fertilization morula stage. IVF_B: in vitro fertilization early blastocyst stage. PA_Z: parthenogenesis activation zygote stage. PA_M: parthenogenesis activation morula stage. PA_B: parthenogenesis activation early blastocyst stage.

## Data Availability

All the RNA-seq data generated in this study have been deposited in the GEO database under the accession number PRJNA595438.

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
