# Peer review of "Transcriptome Analysis of In Vitro Fertilization and Parthenogenesis Activation during Early Embryonic Development in Pigs"

_genes, 2021, doi:10.3390/genes12101461_

Round 1

Reviewer 1 Report

The authors present RNA-seq data on IVF and PA embryos. The methods are detailed, sometimes including unnecessary information. My main concern is that the authors haven't yet decided what their manuscript is about. Is it about the  transcriptional fingerprints of the different developmental stages of PA and IVF embryos or to define the molecular basis for the frequent failure of PA. The results support the former but the introduction and discussion suggest the latter; though not supported by the data. This makes it hard to judge whether the experiments and analyses are adequate. Thus, the introduction and results need to be re-written to focus on what the manuscript is about; transcriptional profiles of IVF and PA embryos. Only then can one reach a logical conclusion about the manuscript. 

The gene expression analyses and presentation are routine, but as stated above, it's hard to judge whether proper comparisons made.

There are several grammatical errors and spelling mistakes that must be corrected e.g., there's nothing like Dseq2. 

Author Response

The authors present RNA-seq data on IVF and PA embryos. The methods are detailed, sometimes including unnecessary information. My main concern is that the authors haven't yet decided what their manuscript is about. Is it about the  transcriptional fingerprints of the different developmental stages of PA and IVF embryos or to define the molecular basis for the frequent failure of PA. The results support the former but the introduction and discussion suggest the latter; though not supported by the data. This makes it hard to judge whether the experiments and analyses are adequate. Thus, the introduction and results need to be re-written to focus on what the manuscript is about; transcriptional profiles of IVF and PA embryos. Only then can one reach a logical conclusion about the manuscript. 

Response: Thank you for your positive comments. It is ture that the methods are detailed, sometimes including unnecessary information. We clearly recognized that the core of our manuscript was about transcriptional changes of the different developmental stages of PA and IVF embryos. Our manuscript described the transcriptional changes caused by the two technologies from the three aspects of mRNAs, lincRNAs and imprinted genes, and in fact, we did not explore the molecular mechanism of these genes using experiments. What we have described or speculated in discussion is based on citing other people’s literature about the molecular basis of these genes. Thus we revised introduction and part of discussion to focus on the transcriptional changes of the different developmental stages of PA and IVF embryos.

The gene expression analyses and presentation are routine, but as stated above, it's hard to judge whether proper comparisons made.

Response: Thank you for your good comments. It is true that The gene expression analyses and presentation are routine. Due to the complexity of horizontal and vertical comparison, the results were separately described in Result 3.2. Differential expression analysis of mRNAs in adjacent stages, IVF and PA. The comparative analysis of adjacent stages in IVF and PA was performed for investigating transcriptional changes of the two technologies’ development processes, and the results were posed in paragraph 1 and 2. Then, comparison in the same stage between the two technologies was performed for observing dynamic differences caused by parthenogenesis activation, which was posed in paragraph 3. Please see lines 288-332.

There are several grammatical errors and spelling mistakes that must be corrected e.g., there's nothing like Dseq2.

Response: We apologized for our mistake. We have replaced “Dseq2” with “DESeq2”. Please see lines 158 and 294. We have corrected the grammatical errors and spelling mistakes, and all revisions are marked in red in the manuscript.

Reviewer 2 Report

The manuscript entitled "Transcriptome analysis of in vitro fertilization and parthenogenesis activation during early embryonic development in pigs" is of major interest in the field of reproduction, especially artificially breeding methods. The authors did a great job and detailly describe their research and critically discuss their results. Therefore, I recommend to publish the manuscript in the current form.

Author Response

Response: Thank you for your good comments on our manuscript!

Reviewer 3 Report

This study indicates that embryos derived from different production techniques have varied in vitro development to the blastocyst stage, and also affect transcription level of corresponding genes, such as imprinted genes.

The study is interesting and well presented. I have only a few minor comments.

Were the oocytes taken from one or more females? Breed, age? Perhaps this could be stated in the Materials and Methods section.

Also, technical editing of the text is necessary.

Author Response

Response: Thank you for your suggestions.

Comments

Were the oocytes taken from one or more females? Breed, age? Perhaps this could be stated in the Materials and Methods section.

Response: Thank you for your good comments. Oocytes used in this study were obtained by in vitro maturation, and the porcine ovaries were obtained from a local slaughterhouse (Please see lines 107-116). Therefore, oocytes used in this study were taken from multiple females and other information like breed and age are unclear.

Also, technical editing of the text is necessary.

Response: Thank you for your good comments and we have tried our best to correct all the grammatical errors. All revisions are marked in red in the manuscript.